# Socioeconomic Conditioning of the Development of the COVID-19 Pandemic and Its Global Spatial Differentiation

**DOI:** 10.3390/ijerph18094802

**Published:** 2021-04-30

**Authors:** Jerzy Bański, Marcin Mazur, Wioletta Kamińska

**Affiliations:** 1Institute of Geography and Spatial Organization, Polish Academy of Sciences, 00-818 Warsaw, Poland; m.mazur@twarda.pan.pl; 2Faculty of Natural Sciences, Institute of Geography and Environmental Sciences, Jan Kochanowski University of Kielce, 25-346 Kielce, Poland; wioletta.kaminska@ujk.edu.pl

**Keywords:** geography of disease, COVID pandemic, socio-economic conditioning, country classification, correlation

## Abstract

The COVID pandemic very quickly became the world’s most serious social and economic problem. This paper’s focus is on the spatial aspect of its spread, with the aims being to point to spatial conditioning underpinning development of the pandemic, and to identify and assess possible socio-economic features exerting an impact on that. Particular attention has been paid to the percentage of positive tests for the presence of the coronavirus, as well as mortality due to the disease it causes. The statistics used relate to 102 countries, with the research for each extending from the time first cases of COVID-19 were reported through to 18 November 2020. The focus of investigation has been the stochastic co-occurrence of both a morbidity index and a mortality index, with intentionally selected socio-economic variables. Results have then been summarized through the classification of countries in relation to the two indices. Highest values relate to Latin America. A significant co-occurrence of morbidity and mortality with GDP per capita has been identified, as values for the indices are found to be lower in wealthier countries. The basic conclusion is that the dependency of the pandemic on environmental and socio-economic conditioning became more complex and ambiguous, while also being displaced gradually as concrete political decisions came to be taken.

## 1. Introduction

COVID-19 is a new global phenomenon affecting all of the world’s countries, albeit to differing degrees. The SARS-CoV-2 coronavirus, which induces a severe respiratory disease, took a mere 2–3 months to become the world’s most serious social and economic problem, with a direct threat to human life posed, and an economic crisis generated that targeted services, trade and culture in particular.

While we are learning more and more about this disease and its virus, we remain unable to cope with its consequences. Nevertheless, the scientific literature does now offer more and more studies whose aim has been to understand at least the causes of spread, with a view to factors favouring the development of the pandemic being identified. These have therefore been studies falling within the scope of geographical analysis—on scales ranging from the global down to the local [1].

One leading example of a global conceptualization of the disease in the literature is a report of research into numbers of cases and death tolls associated with COVID-19 in no fewer than 238 countries—on the basis of WHO data [2]. However, this study is already of a historical nature, given the limited access to data characterized by a hard-to-determine level of reliability, and with what now looks like an “old” date (in the first half of March 2020).

Further research confined itself to a more manageable 79 countries—in different regions of the world, with account taken of the February–May 2020 period. This found a relationship between numbers of fatal cases of COVID-19 and geographical location, as well as regions’ levels of thermal variability [3]. For their part, A. Rodriguez-Pose and Ch. Burlina [4] chose to address the consequences of the pandemic for Europe. Alongside conditioning of a socio-economic nature (GDP per capita, urbanization, outlays on healthcare, population age-structure and level of education), these authors stress the relevance of institutional factors (forms by which decisions are taken and organized, ways in which society is arranged, degree of centralization of power, and so on).

A research-approach taken at the more-detailed level of one megacity (Rio de Janeiro) allowed numbers of cases and fatalities to be assessed in relation to the genders and ages of city inhabitants, with it also proving possible to identify districts most threatened by the development of the epidemic [5].

A broader look at the studies carried out supports assignment of much of the work done on early-stage geographical dimensions of the pandemic to just a small number of main groups [6]. Thus, there are studies engaging in spatio-temporal analysis to account for the pandemic’s scope and consequences [7,8,9], in the sphere of medical geography [10,11], or seeking to determine how far the development of the disease has been dependent on defined kinds of conditioning [12,13]. A further key matter of interest for researchers has related to the methods and instruments deployed in carrying out geographical studies of the type referred to.

Among studies taking account of spatial aspects of the pandemic, the largest group is of those seeking to evaluate the relationship between climatic conditions and the spread of infection [14,15,16,17,18,19,20]. Others have related to the influence of social mobility on the pandemic’s development [21,22,23,24], as well as methods and tools by which geography might analyse the pandemic [25,26,27]. The attention is here drawn to a dramatic increase in accessibility of “Big Data”, allowing for research into the spatial consequences of the pandemic crisis on both the national and global scales [28]. Of course the crisis in question is markedly geographical in nature, inter alia thanks to disruptions to individual mobility.

Here, our focus is on the spatial aspect of the spread of the COVID pandemic, with the aims being to try and point to spatial conditioning underpinning development, as well as to identify and assess possible socio-economic features that have been exerting an impact. Particular attention has therefore been paid to the intensification of infection, in line with numbers of tests for the presence of the coronavirus carried out, as well as mortality due to the disease it causes. Furthermore, our research has focused on the period of occurrence of the first cases of COVID-19 through to the second stage of the pandemic; while diagnostic features taken account of here, as potentially conditioning the development of the pandemic, are density of population, population urbanization, GDP per capita, per-capita outlays on the health service, share of inhabitants aged 65 and over and share of inhabitants with higher education.

## 2. Materials and Methods

The data source Our World in Data [29] is a portal rich in statistical information of very varied and wide-ranging subject matter i.a. developed by Oxford University. Materials on COVID-19 are gathered there and made available by a team comprising several different researchers [30]. The present study makes use of data on numbers of cases noted, of tests carried out for the disease, and of deaths induced by it. It needs to be stressed clearly that these are materials supplied from government-run centres in each country, which all apply different methods of data collection. Inevitably, that has meant diverse data of varying quality, with errors relating to numbers and types of medical tests and examinations that take place.

Our research is, above all, interested in absolute numbers of cases of COVID-19, with reporting on this naturally depending directly on numbers of tests for the presence of the pathogen actually carried out. Difficulties with comparability reflect relative levels of this testing, where two countries have similar populations but quite different levels of testing, and it will need to be clear that the reliability of results is greater where more tests are given. Here, then, is a clear source of possible error or limitation reflecting imperfections in the database, which it has tried to process.

The statistics used here have related to 102 countries making available all of the information referred to above, including the number of tests for the presence of the virus carried out. The research for each country extends from the time first cases of COVID-19 were reported through to 18 November 2020 (the day on which searches through the data in question ceased). Gathered statistical information from the Our World in Data portal was verified by the authors, using data published by the World Health Organization (WHO) in 2020 [31]. WHO’s data on numbers of COVID-19 cases have been published daily since 21 January 2020, while those concerning the death toll have been a constant feature since 2 February 2020. Unfortunately, information on the level of testing for the presence of the pathogen has been lacking.

The approach as described above allowed for the development and calculation of disease morbidity and mortality indices for the countries in question. The index of morbidity ($Wc_{i,j}$) denotes numbers of cases re-expressed in relation to numbers of tests for the presence of the pathogen carried out in country *i* for day *j*. In turn, the COVID-19 mortality index ($Wd_{i,j}$) was calculated by reference to numbers of fatal cases of the disease as related to numbers of infections in country *i* as of day *j*. The construction of the indices was thus as expressed by the formula
(1)$$Wc_{i,j}=\frac{C_{i,j}}{T_{i,j}}$$
as well as
(2)$$Wd_{i,j}=\frac{D_{i,j}}{C_{i,j}}$$
where:

$T_{i,j}$ is the cumulative number of tests for COVID-19 carried out in country *i* from the day of first testing through to day *j* inclusive;

$C_{i,j}$ is the cumulative number of cases of COVID-19 reported in country *i* from the time the first case was noted through to day *j* inclusive;

$D_{i,j}$ is the cumulative death toll due to COVID-19 in country *i* from the time the first death was noted through to day *j* inclusive.

The countries under study were categorized by reference to these three indices. Categories of country with low, average and high morbidity were taken account of, as were countries with low, average or high mortality. The explanation for adopting boundaries for variables of these kinds, by reference to characteristics of a normal distribution, has been supplied just above the equations. In this regard, we have proposed making direct reference to using the following boundaries specifically, i.e., *mean* − 0.431 × *standard deviation* and *average* + 0.431 × *standard deviation*. It therefore needs stressing that this assumption does not imply a necessity for the distribution of empirical data to be near-normal. The aim was rather to identify possible skewing within the set of empirical data, to the point where uneven category sizes would ensue, and it would therefore have been better to seek other ways of matching the real differentiation of the situation present in the set of countries analysed.

Bearing in mind the values for normal-distribution distribuants, a “low” value was taken to be any one whereby $Wc_{i,j}<\bar{Wc_{i,j}}-0.431\sigma \;\mathrm{and}\;Wd_{i,j}<\bar{Wd_{i,j}}-0.431\sigma$. This approach in turn denoted criteria for “high” values with $Wc_{i,j}>\bar{Wc_{i,j}}+0.431\sigma \;\mathrm{and}\;Wd_{i,j}>\bar{Wd_{i,j}}+0.431\sigma$. Once concrete empirical data for 102 analysed countries had been taken account of, the “average” values were deemed to be identified as $Wc_{i,j}\in \langle 4.85\%;17.26\%\rangle$ or $Wd_{i,j}\in \langle 1.38\%;2.65\%\rangle$.

The final effect of the classification activity is a bringing-together of the two categories, allowing each of the 102 countries to be assigned to one of 9 classes (Figure 1). Achieved in this way is a generalized characterization of the given state’s epidemic situation, as contrasted with the remaining items in the set under study.

The indices of morbidity and mortality were also tested (analysed for Pearson linear correlation across the countries) for co-occurrence with selected kinds of socio-economic conditioning. The more sophisticated methods of statistical modelling have been abandoned due to poor data background. Therefore, findings need to be perceived as the initial superficial contribution to possible further in-depth studies. The research question arising was thus as follows: Is there a dependent relationship between COVID-19 morbidity and mortality on the one hand, and on the other certain defined social and economic factors characterizing different countries? The subjects of analysis here were thus variables describing the following potential factors:
$Wse_{1}$—a given country’s population density as of 2018 (people/km^2^) (FAOStat);$Wse_{2}$—the percentage of the given country’s population that was urban as of 2018 (*FAOStat*, *Department of Economic and Social Affairs of the United Nations*);$Wse_{3}$—GDP per capita as of 2017 (USD/inhabitant) (*FAOStat*, *the IMN database*, *CIA Factbook*);$Wse_{4}$—overall annual outlays on healthcare per inhabitant as of 2017 [USD/inhabitant] (*WHO*, *FAOStat*);$Wse_{5}$—the percentage of inhabitants aged 65+ as of 2018 (*Department of Economic and Social Affairs of the United Nations*, *World Bank database*, *CIA Factbook*);$Wse_{6}$—the percentage of the population aged 15 and over in 2010 with higher education [29].

Where two of the six indicators ($Wse_{4}\;\mathrm{and}\;Wse_{6}$) were concerned, the set of countries included in the analysis was limited from the outset (to 88 and 97 respectively) due to limitations with the data. Thus, to ensure reliable results, a further stage of the investigation saw the set of all 102 analysed states limited to those not standing out as having given either an exceptionally small or exceptionally large number of tests per head of population. While this reduced the sample size, it did much to increase reliability and credibility of the study, given the way values for key disease indicators may not be compared in situations where testing is so markedly disparate. On the one hand, a small number of tests done supplies only a small sample of people studied, with the result that the share of positive results is likely to be characterized by major error. On the other hand, there comes a certain point at which a further increase in testing in a given country starts to become prophylactic in nature, and is therefore also encompassing people showing no distinct disease symptoms. The effect then is to lower indices for positive results in comparison with those countries in which numbers of tests as set against population size are smaller.

Empirical data in fact confirm the validity of these suppositions. The set of 102 countries analysed reveals that both morbidity and mortality are correlated significantly with numbers of tests per 1000 inhabitants carried out. The greater the numbers of tests done in given countries, the lower the values of their indices of morbidity and mortality. Ultimately, the analysis therefore precluded 6 countries with the lowest numbers of tests per 1000 inhabitants—i.e., fewer than 10 (which is in turn below the level of 1 test per 100 inhabitants).

## 3. Results

The highest indices for COVID-19 morbidity (Figure 2) are those reported for Latin American countries: Brazil (92%), Peru (83%), Costa Rica (52%), Mexico (45%) and Bolivia (42%). Somewhat further down the list there are other countries from the same region (i.e., Guatemala, the Dominican Republic and Panama). It must be noted that these countries also have very low indices for testing, with the effect that values of the $Wc_{i,j}$ statistic are raised markedly. At the other extreme, values for the index are decidedly the lowest in East Asian countries (China and Vietnam—0.1% and Thailand 0.4%), along with Australia (0.3%), New Zealand (again a mere 0.1%) and Fiji (0.2%).

The situation proves more differentiated when it comes to the index for the mortality COVID-19 induces (Figure 3); but highest values are again noted for Latin American countries (Mexico—9.8%, Ecuador—7.2% and Bolivia—6.25%). Further places in the ranking are taken by Fiji, Iran, China, Peru and Italy. The lowest mortalities associated with the coronavirus are seen to characterize certain countries of Asia (Singapore—0.05%, Qatar—0.2% and the United Arab Emirates—0.35%).

In general, the values for the studied indices averaged by continent serve to confirm the conclusions as above. The morbidity index for the countries of South America exceeded several-fold the values for the analogous index noted for other regions of the world (Table 1). Very high morbidity involving almost 1/5 of people tested for COVID-19 is also noted for countries of North and Central America. In the case of the former, this is above all a reflection of extremely high morbidity in Mexico (at the level of 45.3%). In the other two countries the values are actually below the world average (with the USA on 6.8%, and Canada on 2.9%).

The countries analysed located in the Americas also display relatively high indices for the index of mortality due to COVID-19, these being more than twice as high as the average for the 102 countries taken together (which stands at 2.1%). It is again the North American country of Mexico that reports a very high value for this index (of 9.8%), even as the corresponding figures for the USA and Canada are also rather high (at 3.6 and 2.1% respectively). In turn, Asian, European and African countries as grouped all have mortality index values below the average for the world as a whole.

The classification based on the two indices addressed simultaneously in the matrix shows a majority of countries manifesting average values. Indeed, classes with “low” and “high” values for the indices are less well-represented than would be the case with a normal distribution. This means that, where a given country is assigned to a class of this kind, it is relatively frequent for morbidity or mortality to there depart markedly from the value regarded as “average”. Low values for the indices of morbidity and mortality are present in Europe and Asia above all, while high-value countries (as regards both indices) are mainly located in South America (e.g., Brazil, Peru, Ecuador, Bolivia and Colombia), albeit with Guatemala, Mexico and Tunisia augmenting them (Figure 4, Table 2). It is nevertheless hard to point to any spatial “rules” that would account for the distributions of the classes determined. For example, class *ax* (for which the values of both the $Wc_{i,j}\;\mathrm{and}\;Wd_{i,j}$ indices are low) authentically includes all three Baltic countries (Lithuania, Latvia and Estonia) and even the adjacent Belarus, but then is rounded off by the inclusion of Malaysia, Cyprus and Rwanda. A similar circumstance might apply with class *az* (low $Wc_{i,j}$ together with high $Wd_{i,j}$) whose encompassing of the UK, Ireland and Australia is ostensibly made to look less cohesive by the presence alongside of China, Turkey and Vietnam.

Analyses of the correlations between indices of morbidity and mortality on the one hand and selected kinds of socio-economic conditioning on the other do not attest to the presence of strong dependent relationships, to the extent that values for correlation coefficients did not achieve statistical significance in most cases (Table 3). Only with the level of economic development expressed as GDP per head ($Wse_{3}$) did it prove possible to note a weak relationship, whereby lower morbidity and mortality seemed to be features of wealthier countries, while mortality rates in turn seem to be higher in countries where population density is lower. It is clearly hard to arrive at an unambiguous interpretation of such results, and all the more so as they seem to contradict regularities identified earlier—by which numbers of COVID-19 cases and death tolls did indeed correlate with socio-economic characteristics of the world’s different countries.

Thus, no clear tendencies emerge from a comparison of socio-economic features of the countries allocated to different classes, while conclusions that can be drawn seem surprising to say the least, in some cases (as in the aforementioned case of higher morbidity and mortality apparently being typical for countries with lower densities of population). Equally, a direct comparison of two extreme classes represented by *A−* (with low values for both the $Wc_{i,j}\;\mathrm{and}\;Wd_{i,j}\mathrm{indices}$) and *C+* (with high values for both) suggests that higher morbidity and mortality alike are present in those countries in which the share of people in the 65+ age group is only just over half as high (Table 4). On the other hand, *A−* countries have values for GDP per capita and outlays on the health service that are several times higher (respectively of 32,190 and 6135, as well as 116.6 and 23.3).

## 4. Discussion

There was a rather long period over which numbers of countries reporting COVID-19 infections remained low. For any first clear change in number we must look to late January, at which point the 25th country of occurrence had been noted. Back then, it was first and foremost states in just two categories that were still involved, i.e., ones highly-developed from the economic point of view (USA, Germany, France, Italy, Australia, Sweden, Canada, etc.) or else neighbours of China itself (including Cambodia, The Philippines, Nepal and Russia). After around 20 days there was a further, abrupt increase in numbers of countries in which infections had been recorded, but it remains hard to suggest why a relatively prolonged stabilisation—apparent or real—might have been present. Most likely there are several factors involved, including simply the lack of testing to reveal the real situation, a tendency for governments and local authorities to play things down and relatively lengthy periods of incubation of the disease. It was thus only in late February and early March 2020 that the curve for the numbers of countries with reported cases really began to change, with the increase thereafter tending to proceed steadily to the point where almost everywhere in the world had been reached by COVID-19.

The first period of transmission of the novel coronavirus was most probably associated with mobility of the population and its concentration in large urban centres. First cases of COVID-19 infection were thus noted (and probably took place) in rich regions of the world. The later development of the pandemic was more complex, with morbidity and mortality most likely coming to depend on interactions between a wide variety of causes and types of conditioning. To be distinguished among these is biological (including genetic and immunological) conditioning, other kinds of conditioning present in nature (e.g., relating to air temperature and humidity, insolation, and specifics of the flora and fauna), political conditioning (as regards remedial measures of differing spatial scope) and conditioning of a socio-economic nature (e.g., relating to levels of economic development, public awareness, the state of the health service, cultural factors and lifestyle).

The key and sometimes dominant role of such conditioning when it comes to both the spread and different consequences of infectious diseases can be seen by reference to history’s worst and most widely fatal epidemics and pandemics. Exemplifying biological conditioning—and notably unprepared immune systems among communities living in isolation for centuries—are the tragic consequences and impacts of new discoveries made by explorers and then followed by first colonisations. Absolute swathes were cut through the populations encountered at this stage. In 1520, smallpox was brought to the Yucatan coast by the Spanish fleet, only to go on and kill so many that population on the territory of today’s Mexico most likely declined from 22 to 14 million [32]. A smaller absolute impact, but far greater in relative terms, was exerted over around 75 years following the discovery of Hawaii by Britain’s Captain James Cook (in 1778). That contact between civilisations spreading influenza and chickenpox viruses, as well as the bacteria inducing TB, syphilis and typhus, resulted in a decline in numbers of inhabitants of the island chain from 500,000 to 70,000 [33].

Socio-economic factors have also had a very major influence in accounting for differences in the spread of infectious diseases. This can be traced in respect of the 14th-century epidemics of the bacterial disease known simply as The Plague or Black Death. This most likely originated in China, and it spread in relation to two vectors in the form of fleas, as well as the black rats in whose fur they resided and could be transferred. Such a role in transmission favoured the spread of the disease along the Silk Road/Routes and into Europe, where perhaps 30% or even 60% of the then population succumbed [34]. However, relative death tolls were still highest in the areas most developed economically and populated most densely [35].

Far more recently, it was very much a matter of the state of a country’s health service, as well as the knowledge and awareness of its society, that helped determine how fast and effectively the viral disease AIDS might spread and cause deaths [36]. WHO figures reveal that AIDS killed around 605,000 people in 2018 alone, with as many as 472,000 of these deaths among inhabitants of Africa, as opposed to just 11,400 in Europe, where the level of development of medical care is much higher.

A particular intensity to the present pandemic is revealed in the values for indices of morbidity and mortality obtained for Latin American countries. However, to that should be added an observation regarding the relatively low values for testing in the region, generally below 40 tests per 1000 inhabitants (as compared with the mean for all countries studied at 224). This in the face of the clear relationship noted for the dataset as a whole, whereby countries in which few tests have been carried out do manifest higher levels of morbidity and mortality. Basically, testing in Latin America mainly revolves around those showing clear symptoms of having been infected.

It is worth noting that almost all of the countries in which the $Wc_{i,j}$ index assumes highest values are in the tropics, even as the countries with low values for this index are in the temperate zone, and in particular located at higher latitudes. This might point to a significant influence of climatic conditioning where coronavirus is concerned. However, to date there has been a lack of research confirming unequivocally that this factor has an influence, despite study on climatic conditioning of the disease being among the most abundant within the subject literature [14,16,19]. Studies conducted by a team of Brazilian academics have made it clear that any statements to the effect that COVID-19 spread less rapidly in warmer and more humid countries need to be treated with a great deal of caution, given the anecdotal nature of the claim and lack of evidence for it [37]. Our results show how those fears look justified, as all states included in class *C+* (with the highest values for the $Wc_{i,j}\;\mathrm{and}\;Wd_{i,j}$ indices) are indeed located in zones of tropical or subtropical climates.

On the other hand, the influence of natural factors overlays socioeconomic impacts. Given the high dynamics and short duration of the phenomenon, available data do not allow for separate analysis of these two groups of conditioning factors, e.g., with comparisons of values for the $Wc_{i,j}\;\mathrm{and}\;Wd_{i,j}$ indices in different climatic zones in line with an assumption that comparable socio-economic conditioning is present there. In the context of the present study, these kinds of conditioning did not manifest strong linkage with either the development of the pandemic or its consequences. While it would appear that a dataset comprising some 100 studied units and with the time scope that was present (including each day from the time of first recording of an infection through to November 2020) would offer a basis for obtaining objective results, the fact is that just three of our 12 analysed relationships achieved statistical significance.

What the data inter alia demonstrated was that levels of COVID-19 morbidity and mortality are higher where the GDP per capita index is lower. However, even this relationship is most probably linked with other kinds of co-occurring conditioning that relate not least to outlays on healthcare, lower levels of awareness in society, more-limited opportunities to obtain information on the disease and, of course, less favourable existential conditions (in particular of a sanitary nature).

The lack of statistically significant correlations may not necessarily mean that relationships are lacking altogether, but simply that the sample available did not provide for these to be demonstrated. Moreover, the construction of the $Wc_{i,j}$ and $Wd_{i,j}$ indices has also had a major impact on the results. The former in particular (as related to numbers of tests), beside potential reasons listed above, is strongly exposed to the impact of the current state of the pandemic diagnosis within a given country. The more people tested preventively, the lower the share of morbidity evidenced. In many cases a good diagnosis is an obvious effect of high economic availabilities of national healthcare. Such an approach was taken intentionally, in line with an assumption that the actual state of the pandemic is better represented by means of a relationship with the number of tests as opposed to population size. The opposite approach is applied in the literature quite often. However, it is worth emphasizing that a very poor state of diagnosis in a number of countries, in the current state of the pandemic, implies a positive impact on the COVID statistics there.

It is clear that these opposite approaches lead to different results, as is evidenced in some of the literature. For instance, earlier research by the same authors concerned itself with the first (spring 2020) phase of the pandemic, and had a converse relationship to offer, as COVID-19 morbidity was found to be higher where countries’ levels of urbanisation and economic development (GDP per capita) were higher. Furthermore, a still-clearer relationship was found with expenditure on the health service per inhabitant [38]. That research was concerned with a shorter period, but also with a larger and more evenly distributed set of countries. There is, thus, every reason to attach equal weight to its findings as to those in the present research. While it is true that the effect detected was most likely due in part to the high domestic and international mobility of the inhabitants of rich countries, as well as more effective diagnostic techniques and mechanisms, there may nevertheless be a “rule” applying overall here, when it comes to the conditioning of COVID-19’s spread.

The first cases of coronavirus outside Asia were noted on 23 January 2020 in the United States, and in Australia and France on 25 January 2020. In contrast, in Africa there was no reported case until 15 February 2020 (and that was in the northern part, in Egypt), while in South America (Brazil) the disease situation began to develop from 27 February 2020. However, it is not known how or to what extent the distinct “time lag” apparently applying to Africa and South America, as opposed to other parts of the world, reflects their distance and relatively weaker international linkages (in trade, migration, tourism and so on), as opposed to more limited achievements in diagnostic medicine that also reflect more limited development of health services overall.

The linkage between socio-economic factors and levels of COVID-19 morbidity and mortality would thus seem to relate to the phase of the pandemic. Research should therefore draw a clear distinction between the first phase and later ones, given that these seem to have been shaped by different conditioning.

The authors’ studies as detailed here were concerned with a relatively large number of countries located in different parts of the world, as well as a quite lengthy time period, linked at least to the COVID-19 pandemic’s second phase of development. It could thus be concluded that the statistical material gathered did offer a more objective assessment of the pandemic’s development and consequences than did analysis carried out in the earlier first period.

It nevertheless emerges that the socio-economic conditioning of the pandemic detected successfully by the authors in the earlier phase was both more distinct and more convincing. Our analysis of the dynamics of numbers of infections in the first phase of the pandemic took account of 162 countries and showed how the highest values for this had been present in Europe [38]. Overall, among 20 countries in which the increase in numbers of cases during the first 15 days following first identification was more than 100-fold, no fewer than 15 of these were European (e.g., Denmark with a 413-fold increase in numbers of infected in the first 15 days of the epidemic nationally, Slovenia—319-fold, Sweden—310-fold, Spain—247-fold and Portugal—224-fold), along with 4 Asian states and just 1 in Central America.

Dynamic increases in numbers of infections in the first phase of the pandemic were inter alia characterised by significant statistical linkage with urbanisation. The higher the level of the latter, the higher the COVID morbidity to be noted in the given country. However, our new analysis relating to the “mature” stage of the pandemic has failed to confirm a dependent relationship between the $Wc_{i,j}\;\mathrm{and}\;Wd_{i,j}$ indices and urbanisation, thus contrasting and being at odds with the results of earlier studies. Coelho et al. [37] showed how it was precisely the large, densely populated and highly accessible cities (London, Paris, Milan and Madrid) that were places to which and through which the disease spread during the initial phase of the pandemic. Indeed, its highest rates of development were characteristic of the cities serving as global transport hubs [39]. Analogous conclusions were reached by R. Florida [40], who identified three groups of areas of concentration of the virus, i.e., urban “superstars” attracting huge influxes of tourists and other travellers (like London and New York), as well as industrial centres linked via supply chains (e.g., Detroit and northern Italy), and magnets for tourists (e.g., Italy, France and Switzerland).

High values for the level of urbanisation and population density point to situations that naturally favour rapid infection of a larger number of inhabitants, in the face of near-insurmountable difficulties with avoiding direct contacts with others. Indeed, urbanised, densely populated areas usually—for those very reasons—utilise public transport, large shops and shopping centres, as well as public institutions dealing (designed to deal) with large numbers of people. Furthermore, a great part of the urban population even resides in circumstances of multi-family housing.

However, the above remarks and examples would all seem to apply particularly to the pandemic’s early phase, while relating to a fine scale (of the selected urban centre or region). Later restrictions on air traffic and other measures to limit freedom of movement were in theory—and it seems in practice—enough to ensure that urbanisation ceased to serve as a key indicator where the spread of the SARS-CoV-2 virus and COVID-19 were concerned.

It was also impossible to demonstrate any strong connections between the development of the pandemic and population density. Indeed, a paradoxical finding was that values for the mortality index actually seem somewhat higher in countries whose densities of population are lower. The role played by population density in the spread of COVID has indeed been questioned, it in particular being noted that the influence is only greater where early stages of the disease outbreak are concerned [4,41,42].

It also needs emphasizing how the analysis carried out has failed to confirm a relationship between COVID-19 morbidity or mortality and levels of financial outlays on health services. This was noteworthy, given that the same authors’ earlier research—into the pandemic’s first phase—did identify a highest level of correlation with outlays on the health service [38]. The contention is that, in practice, countries enjoying the highest quality of life facilitate preventative activity via their highly developed healthcare, even as this usually fails to compensate entirely for the impact of rich-country lifestyle factors doing so much to stimulate COVID’s dynamic spread.

Beyond that, a risky thesis that might be advanced is that a high level of healthcare (as confirmed in levels of outlays on health services) may actually have had some negative effect in the pandemic’s growth phase, through reinforcement of the trend for numbers of cases to increase. A lack of adequate knowledge on this coronavirus and vulnerability to infection by it may have combined with initially inadequate equipping of health services in PPE, as well as influxes of infected people into a large number of healthcare establishments, to encourage transmission of the pathogen. Non-theoretical examples of this were of course provided by care homes, at which any appearance of the virus had the subsequent effect of infecting large numbers of people there, be they either the cared-for or the carers.

A further, in fact somewhat surprising, result is the lack of a relationship between countries’ COVID-19 morbidity and mortality on the one hand and their shares of older people within the population age structure. Both the media and the scientific literature seem to have made universal reference to the “rule” that elderly people are more vulnerable to infection and more likely to die [43,44,45]. However, data presented usually relate to highly developed countries (mainly European), in which the age structure of inhabitants features a very high share of older people. Overall, in this present study it is Asian and African countries that manage to amount to a major component, and in these age structures populations continue to be more in the classic pyramid shape, meaning a decidedly smaller share of the population is accounted for by the 65+ group. This may actually reach the point at which the impact of this group of older people on the figures becomes “vanishingly” small when set against inhabitants in all the other age groups.

Meanwhile, at a more-detailed level, our analysis shows how the 20 countries in which the share of the 65+ group in the population is highest include only 4 in which values for the index of COVID mortality are above the 2.01% mean noted for the world as a whole, or at least the 102 countries we selected to represent it. The countries in question are Italy—3.75%; Spain—2.76%; Bulgaria—2.26% and Hungary—2.15%. These findings make it worth re-emphasizing that the present authors’ earlier research—on COVID-19 mortality through to 5th April 2020 in 44 countries (more than half of these in Europe)—also failed to demonstrate a statistically significant effect on mortality that could be associated with the share of older people in the population. Thus, high mortality due to COVID-19 may only partially be accounted for by an “old” demographic structure of the population.

Of course, where given individual sufferers are concerned, the data on COVID-19 mortality at different ages confirm this factor’s significance in an absolutely clear way, leaving it impossible to claim that age structure will have no effect on mortality. However, the influence would seem much easier to register on the micro-scale, when we look at differences in mortality between social units at some core level, or else local spatial units. Where a larger spatial scale (regional, national or international) is involved, the search for differences in mortality sees population age structure relegated to just one of many key factors. When account is also taken of, say different countries’ varied cultural norms, the particular pandemic-related decisions their administrations have chosen to take, or differences in levels of discipline and respect for the rules from one society to another, it becomes far more difficult to say that there is greater mortality precisely where senior citizens constitute a larger share of the national population. However, there is still no gainsaying of the thesis that, in each given country featuring its particular definable conditioning, mortality would still be more limited were the society to be younger overall.

## 5. Conclusions

As time has passed and the disease has spread further throughout the world, the process of intensification and the consequences of that have come to depend on a wider variety of factors, to the point where a system of complex interrelationships has taken shape. It is seemingly for this reason that individual key aspects to the conditioning fail to reveal significant linkage with the researched features of the disease. Ultimately, the analyses run by us were only able to associate greater COVID-19 morbidity and mortality with countries in which GDP per capita is lower, as well as revealing a link by which higher mortality characterises countries whose population densities are lower.

The scale of the research also impacts upon its results. Here, each country—irrespective of size, location and numbers of people—is treated as an equivalent statistical unit. It seems probable that more detailed analysis based around regions more similar in their geographical features would have yielded more concrete results. It is possible to note a regularity by which countries with a low or average level of morbidity are associated with mortality not dependent on age structure, while countries characterised by a high level of morbidity experience mortality at a higher rate where the shares of people aged 65 and over are lower.

The very general conclusion emerging from our research is that the development of the COVID-19 pandemic displays major spatial differentiation as regards conditioning. This means that global-scale analyses of the issue must be burdened by far-reaching limitations on their results. It is also possible to hypothesise that, as the time over which it has persisted grows ever-longer, the spatial differentiation of the pandemic’s consequences comes to depend more and more on concrete political and administrative decisions taken, as well as on the timing of their implementation. To an ever-greater degree these may allow for the evening-out of possible influences due to the unfavourable natural and socioeconomic conditioning potentially present in different countries.

## Figures and Tables

**Figure 1 ijerph-18-04802-f001:**
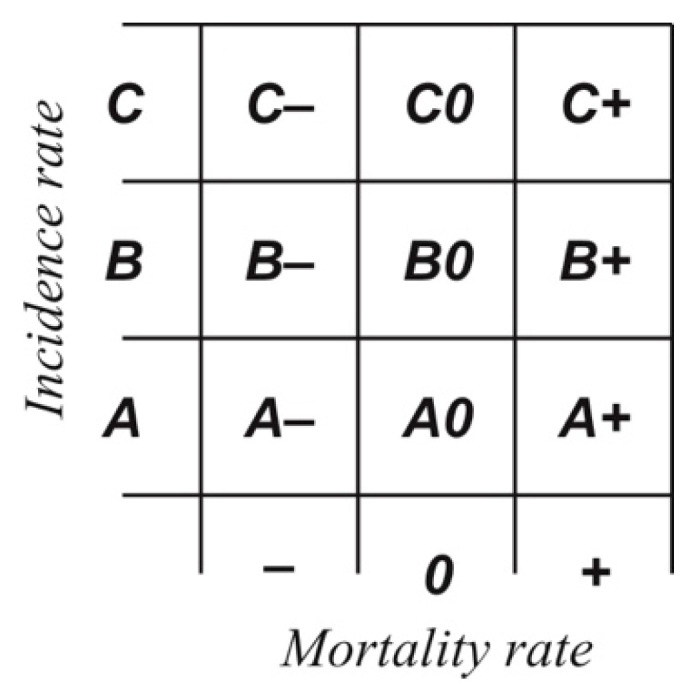
The matrix of classes accounting for both levels of infection with COVID-19 and mortality rates among those the pathogen infected identifies: A−, low-morbidity/low-mortality countries where COVID-19 is concerned; A0, low-morbidity/average-mortality countries; A+, low-morbidity/high-mortality countries, B−, average-morbidity/low-mortality countries; B0, average-morbidity/average-mortality countries; B+, average-morbidity/high-mortality countries, C−, high-morbidity/low-mortality countries; C0, high-morbidity/average-mortality countries; C+, high-morbidity/high-mortality countries..

**Figure 2 ijerph-18-04802-f002:**
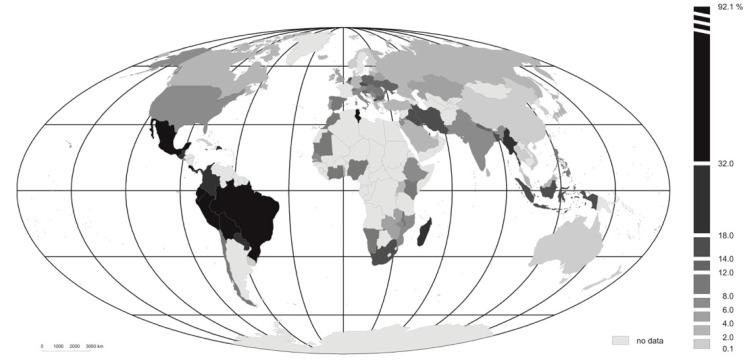
The COVID-19 morbidity index ($Wc_{i,j}$) in different countries, in the period from the first recorded case of the disease through to 18 November 2020. Source: authors’ own elaboration.

**Figure 3 ijerph-18-04802-f003:**
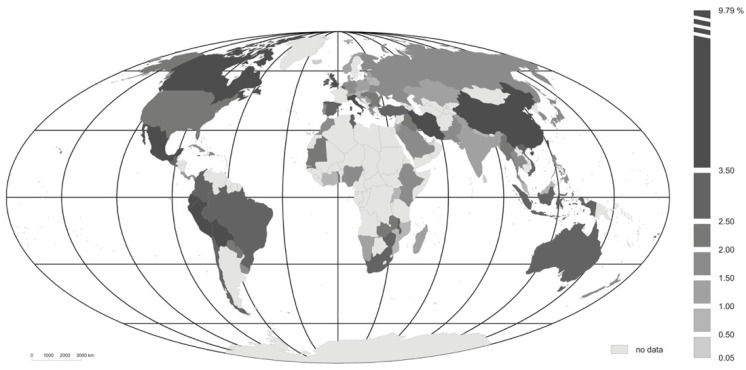
COVID-19 mortality index ($Wd_{i,j}$) in different countries, in the period from the first recorded case of the disease through to 18 November 2020. Source: authors’ own elaboration.

**Figure 4 ijerph-18-04802-f004:**
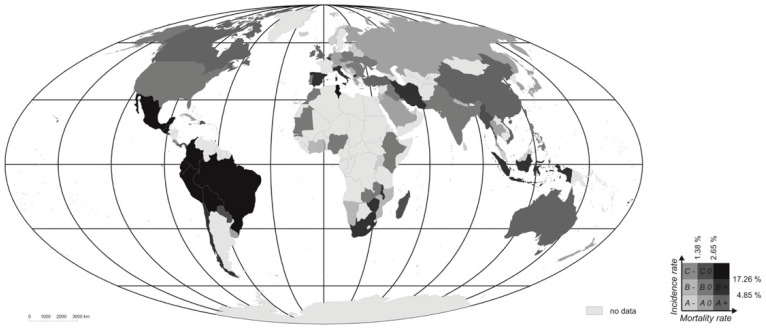
The classification of countries by indices of both COVID-19 morbidity and mortality—in the period from the first reported cases of the disease through to 18 November 2020; A−, low-morbidity/low-mortality countries where COVID-19 is concerned; A0, low-morbidity/average-mortality countries; A+, low-morbidity/high-mortality countries, B−, average-morbidity/low-mortality countries; B0, average-morbidity/average-mortality countries; B+, average-morbidity/high-mortality countries, C−, high-morbidity/low-mortality countries; C0, high-morbidity/average-mortality countries; C+, high-morbidity/high-mortality countries. Source: authors’ own elaboration.

**Table 1 ijerph-18-04802-t001:** Indices for COVID-19 morbidity and mortality in the analysed countries as broken down by continent.

Continent	No. of Countries Studied	Morbidity Index	Mortality Index
Africa	20	9.3	1.7
Central America	8	18.9	2.1
South America	8	43.4	4.0
North America	3	18.4	5.2
Australia and Oceania	3	0.2	3.5
Asia	27	7.2	1.5
Europe	34	7.1	1.6

Source: authors’ own elaboration.

**Table 2 ijerph-18-04802-t002:** The studied countries as classified in terms of both COVID-19 morbidity and mortality among those infected.

Details		Mortality		
		Low	Average	High
**Morbidity**	**High**	Costa Rica	Trinidad and Tobago, Panama, Dominican Republic, Paraguay, Myanmar, Madagascar	Brazil, Colombia, Tunisia, Peru, Mexico, Bolivia, Ecuador, Guatemala
	**Average**	Croatia, Serbia, Slovenia, Austria, Switzerland, Slovakia, Israel, Nepal, Jordan, Maldives, Namibia, Ghana, Mozambique, Cote d’Ivoire, Kuwait, Qatar	Portugal, Bulgaria, Czechia, Netherlands, Hungary, Romania, Poland, Ukraine, United States of America, Jamaica, Morocco, India, Bangladesh, Philippines, Pakistan, Ethiopia, Iraq, Mauritania, Senegal, Nigeria, Kenya, Zambia	Italy, Spain, Belgium, North Macedonia, Chile, El Salvador, Iran, Indonesia, South Africa, Zimbabwe, Malawi
	**Low**	Malta, Latvia, Denmark, Lithuania, Estonia, Norway, Belarus, Iceland, Luxembourg, Cyprus, Singapore, Sri Lanka, Malaysia, Rwanda, Bahrain, Uganda, United Arab Emirates	Japan, Finland, Greece, Germany, New Zealand, Cuba, Uruguay, Russia, South Korea, Thailand, Kazakhstan, Saudi Arabia, Togo	United Kingdom, Canada, Australia, Ireland, China, Turkey, Vietnam, Fiji

Source: authors’ own elaboration.

**Table 3 ijerph-18-04802-t003:** Relationships (expressed in terms of correlation-coefficient values) between the indices under study and selected kinds of socio-economic conditioning. Statistically significant values are bolded.

Details	Density of Population	Population Urbanization	GDP * Per Capita	Per-Capita Outlays on the Health Service	Share of Inhabitants Aged 65 and Over	Share of Inhabitants with Higher Education
**Morbidity index**	−0.105	0.102	**−0.228**	−0.183	−0.171	−0.108
**Mortality index**	**−0.210**	0.038	**−0.209**	−0.177	−0.086	−0.780

* Gross Domestic Product shared by the average yearly number of populations for a given country. Source: authors’ own elaboration.

**Table 4 ijerph-18-04802-t004:** Mean values of the indices noted for the 9 identified categories of countries.

Classes	Density of Population	Population Urbanization	GDP per Capita	Per-Capita Outlays on Healthcare	Share of Inhabitants in the 65+ Age Group	Share of Inhabitants with Higher Education	Morbidity Index	Mortality Index
***A−***	794.1	71.2	32,190	116.6	12.7	12.0	2.2	0.7
***A0***	128.2	75.9	22,531	45.1	14.8	14.4	2.1	2.1
***A+***	121.8	67.6	29,527	120.9	12.2	12.7	1.8	3.7
***B−***	240.8	61.8	21,353	95.8	9.7	7.1	9.7	0.8
***B0***	204.9	53.9	10,209	44.2	10.2	7.7	10.0	1.8
***B+***	143.2	64.9	13,131	19.9	10.7	6.9	10.7	3.2
***C−***	97.9	78.6	11,514	30.7	9.5	14.7	51.8	1.3
***C0***	115.3	55.3	7306	26.4	6.8	6.4	55.5	1.9
***C+***	59.0	72.8	6135	23.3	7.5	8.4	36.1	4.9

Source: authors’ own elaboration.

## Data Availability

Data are available in this article.

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
