# Peer review of "Socioeconomic Conditioning of the Development of the COVID-19 Pandemic and Its Global Spatial Differentiation"

_ijerph, 2021, doi:10.3390/ijerph18094802_

Round 1

Reviewer 1 Report

The paper is valuable contribution to geographical exploring of pandemic Covid-19 global spreading. Title, abstract, key words are accurate and according to use of academic writing. Authors have tried to check spatial distribution of two indexes: morbidity and mortality of pandemic Covid-19 in more than 200 countries all over the world. The research of it is well designed, because help to examine extreme indexes (both law or high) or their combinations (for example low mortality and high morbidity). This study in global scale consists of really big data of the first year of pandemic. Statistical methods are based on international statistics and they are adequate to the aim of the analysis. Both indexes (morbidity and mortality) are well chosen. Conclusions are proper and clear. References demonstrate good authors’ orientation in the newest literature.

The strongest point of this paper is its topic, because all questions of Covid-19 pandemic are now, and they will be in close future crucial for many scientific disciplines including geography as a science with a space as a main object. Big value is also a global scale of the study. Authors have not reduced their attention to main two indexes only, but they have examined also territorial context (density of population, urbanisation, GDP per capita, per capita outlays to health service and share of elderly people and educated people on the whole population of the area). They are informative data that can explain a lot, but no distinctive dependences were examined, at most vague ones. But it is a valid result, undoubtfully.

Weak point of this paper is only one. It is a lack of discussion of the data reliability reporting morbidity and mortality by different countries. There is very high probability of incomplete reporting of both main indexes especially in the less developed countries. Differences are even among developed countries. Some of them (for example Singapore) report death only if Covid-19 is unambiguously case of the death but the others report any suspicious death as connected with Covid-19 (for example Belgium). Difference between Singapore and Belgium according reported data is enormous, but in fact it should be perhaps smaller.    

The global study ought to be general, of course, because such differences are not so important in big data scale. But it would be better to read in this text at least a paragraph with statement that authors are aware of this problem.

Author Response

Thank you for your positive feedback on our study. We agree with the critical remark concerning the problems arising from the quality of the source materials. We have made the additions suggested by the Reviewer.

What this means inevitably is diverse data of varying quality, and burdened by error relating to numbers and types of medical tests and examinations that take place. Our work is above all interested in the absolute numbers of cases of COVID-19, with reporting on this naturally depending directly on the numbers of tests for the presence of the pathogen that are actually carried out. Difficulties with comparability reflect relative levels of this testing – where two countries have similar populations but quite different levels of testing, it will need to be clear that the reliability of results is greater where more tests are given. Here, then, is a clear source of possible error or limitation reflecting imperfections in the database which it has tried to process.”

Reviewer 2 Report

The authors describe morbidity and mortality indices for 102 countries classifying these as low, medium, high on these two aspects.
The authors try also to highlight associations between these two aspects and some characteristics of the country such as the density of population, GDP per capita, per capita outlays on the healthcare, share of inhabitant aged 65 and over, and share of inhabitant with higher education.

Major comment
The findings are very poor; probably some characteristics present a different association with morbidity and mortality indices over the different countries. The authors have to fit a model.

Minor comments
- Figures and Tables are not called on the text. I think that the reader could be helped to follow the research paper by including the references.

- When values are taken on page 3 (lines 126-128) for low, high, and average, explain the use of quantile 0.431.

- Figure 2, Figure 3, Figure 4 are not so clear; a simple rectangular map, with gray-shaded colors, white for the sea, could be better readable.

- Figure 5 perhaps is unnecessary. 

Author Response

Major comment

The findings are very poor; probably some characteristics present a different association with morbidity and mortality indices over the different countries. The authors have to fit a model.

The comment "the findings are very poor" is incomprehensible and enigmatic. We have obtained results which have been discussed and debated. In the analysis, we selected characteristics that we thought would have strong associations with COVID morbidity and mortality levels. We also had a limited resource of materials and statistical features. However, testing for correlations proved to be 'weak' or statistically insignificant. All countries are analyzed under the same assumptions. Interference by the authors taking into account "some" differences between countries could introduce serious errors.

Minor comments
- Figures and Tables are not called on the text. I think that the reader could be helped to follow the research paper by including the references.

As suggested by the reviewer, references to figures and tables have been introduced

Row 148, 202, 214, 227, 250, 268, 288,

- When values are taken on page 3 (lines 126-128) for low, high, and average, explain the use of quantile 0.431.

The explanation of taking such boundaries of variables, referring to characteristics of normal distribution, has been delivered just above the equations. Regarding this remark, we have proposed for more direct referring to using exactly such boundaries, namely: average â€‘ 0,431 * standard deviation and average + 0,431 * standard deviation.

- Figure 2, Figure 3, Figure 4 are not so clear; a simple rectangular map, with gray-shaded colors, white for the sea, could be better readable.

The remark has been accepted. New version of maps have been added. Row 210, 219, 258, Changed the colour scheme of the maps according to the reviewer's proposal.

- Figure 5 perhaps is unnecessary. 

Row 309. Deleted figure. We agree with the reviewer's opinion that Figure 5 is superfluous, because it is not an important element referring to the discussed study results.

Reviewer 3 Report

Comments to “Socioeconomic conditioning of the development of the COVID-19 pandemic and its global spatial differentiation”:

- Lines 27-30, needs to be cited.
- The term work in science is not advisable, use some synonym in this field, as a manuscript
- It is not clear in the introductory section what is the general and explicit objective you are pursuing
- Add the structure of the manuscript
- Can you include other works that have used this method?
- "Categories of country with low, average and high morbidity were taken account of, as were countries with low, average or high mortality". What literature supports these categories?
- Includes a section prior to conclusion, with the main discussions of the results obtained in relation to the literature used
- An excessive use is made of the personal form (we) in the writing. The scientific article must overcome this barrier and be written in an impersonal way.

Author Response

- Lines 27-30, needs to be cited.

The first paragraph of the study is so general and obvious that it does not require proof. In our opinion, it is not necessary to confirm with literature the fact that the pandemic has brought about changes in social and economic life all over the world.

- The term work in science is not advisable, use some synonym in this field, as a manuscript

The term 'work' has a wider connotation than the technical term 'manuscript' referring to a specific text. Moreover, our text was translated by a native-speaker with 25 years of experience in working on geography texts.

- It is not clear in the introductory section what is the general and explicit objective you are pursuing

An addition has been made and now the purpose and focus of the study is:

Here, our focus is on the spatial aspect of the spread of the COVID pandemic, with the aims being to try and point to spatial conditioning underpinning development, as well as to identify and assess possible socio-economic features that have been exerting an impact. To these ends, particular attention has been paid to the intensification of infection, in line with numbers of tests for the presence of the coronavirus carried out, as well as of course mortality due to the disease it causes. The research focuses on the period of occurrence of the first cases of COVID-19 to the second stage of the pandemic. Diagnostic features taken account of as potentially conditioning the development of pandemic were: density of population, population urbanization, GDP per capita, per capita outlays on the health service, share of inhabitants aged 65 and over, share of inhabitants with higher education.

- Add the structure of the manuscript

The structure of the manuscript is in our opinion correct: Introduction, Materials and Methods, Results, Discussion, Conclusions

- Can you include other works that have used this method?

The methodology of the study is author's own and was not taken from another study

- "Categories of country with low, average and high morbidity were taken account of, as were countries with low, average or high mortality". What literature supports these categories?

The categories of countries do not derive from the literature but from the assumptions of the method used by the authors (see Fig. 1 and the limit values of the indicators )

- Includes a section prior to conclusion, with the main discussions of the results obtained in relation to the literature used

Included section Discussion

- An excessive use is made of the personal form (we) in the writing. The scientific article must overcome this barrier and be written in an impersonal way.

The term "we" (used 4 times) has a different context and does not refer directly to the authors.

Round 2

Reviewer 2 Report

My minor comments have been solved with respect to the previous version of the paper. 
Nevertheless, according to me, my major criticisms still be present.

Author Response

The principal aim of the procedure was to gain a spatial pattern of COVID’s morbidity and mortality across the countries and to initially recognize their possible association with some basic socio-economic conditions, to as broad spatial extent and sample of countries to be studied as possible. The basic assumptions of the paper were that: the spatial dimension of the research is a priority and comparability of the results needs to be ensured. Unfortunately, data coverage occurred to be a strong limitation in this regard. It was a reason, why a broad spatial extent of comparable statistical data has been achieved at the expanse of in-depth statistical representation of socio-economic characteristics of the countries. It forced us to keep a paper at the level of superficial contribution to the recognition of possible association (but at relatively close to worldwide range). Therefore also building a model taking into account such small number variables and their possible cross-referencing was unjustified. The short explanation for such reasoning has been added on the page 4.

Reviewer 3 Report

Congratulations for the effort.

Author Response

Thank you. I have added a few sentences on methodology.